# Therapeutic Potential of Pretreatment with Exosomes Derived from Stem Cells from the Apical Papilla against Cisplatin-Induced Acute Kidney Injury

**DOI:** 10.3390/ijms23105721

**Published:** 2022-05-20

**Authors:** Te-Yang Huang, Miao-San Chien, Wen-Ta Su

**Affiliations:** 1Department of Orthopedic Surgery, Mackay Memorial Hospital, Taipei 104217, Taiwan; haungt33@gmail.com; 2Department of Chemical Engineering and Biotechnology, National Taipei University of Technology, Taipei 104217, Taiwan; cf1811@ntut.edu.tw

**Keywords:** stem cells from the apical papilla, exosome, cisplatin, acute kidney injury, oxidation and apoptosis

## Abstract

Acute kidney injury (AKI) is the most serious side effect of treatment with cisplatin in clinical practice. The aim of this study was to investigate the therapeutic effect of exosomes derived from stem cells from the apical papilla (SCAPs) on AKI. The medium from a SCAP culture was collected after 2 d of culture. From this, SCAP-derived exosomes (SCAP-ex), which were round (diameter: 30–150 nm) and expressed the characteristic proteins CD63 and CD81, were collected via differential ultracentrifugation. Rat renal epithelial cells (NRK-52E) were pretreated with SCAP-ex for 30 min and subsequently treated with cisplatin to induce acute injury. The extent of oxidative stress, inflammation, and apoptosis were used to evaluate the therapeutic effect of SCAP-ex against cisplatin-induced nephrotoxicity. The viability assay showed that the survival of damaged cells increased from 65% to 89%. The levels of reactive oxygen species decreased from 176% to 123%. The glutathione content increased by 78%, whereas the levels of malondialdehyde and tumor necrosis factor alpha (TNF-α) decreased by 35% and 9%, respectively. These results showed that SCAP-ex can retard oxidative stimulation in damaged kidney cells. Quantitative reverse transcription–polymerase chain-reaction gene analysis showed that they can also reduce the expression of nuclear factor-κβ (NF-κβ), interleukin-1β (IL-1β), and p53 in AKI. Further, they increased the gene expression of antiapoptotic factor B-cell lymphoma-2 (Bcl-2), whereas they reduced that of proapoptotic factors Bcl-2-associated X (Bax) and caspase-8 (CASP8), CASP9, and CASP3, thereby reducing the risk of cell apoptosis.

## 1. Introduction

Acute kidney injury (AKI) is a clinical condition characterized by a rapid decline in renal excretion function within hours or days, as well as the accumulation of nitrogen-containing metabolites (e.g., creatinine and urea) and other clinically unmeasurable metabolic waste. Other common clinical manifestations include reduced urine output, accumulation of metabolic acids, and increased potassium and phosphate concentrations [1,2]. There are many causes of AKI, including kidney surgery, medication, and sepsis. Nephrotoxic drugs are an important cause of AKI. Severe AKI is associated with uremia, which results in the deterioration of kidney function and other organs throughout the body. These effects may lead to chronic kidney disease, a permanent requirement for hemodialysis, or death. There are many commonly prescribed drugs (e.g., aminoglycosides, angiotensin-converting enzyme inhibitors, and calcineurin enzyme inhibitors, nonsteroidal anti-inflammatory drugs) for this situation [3]. Losartan, a selective competitive angiotensin type II receptor antagonist, reduces the risk of progression to chronic kidney disease and death [4]. In this study, we used losartan as a positive control for comparison with the therapeutic effect of SCAP-ex.

Cisplatin is a platinum-containing antineoplastic agent that is effective against a variety of human tumors, including bladder, head and neck, lung, ovarian, and testicular cancer [5]. Cisplatin forms coordination bonds with DNA bases, thereby deforming the DNA structure. This leads to inhibition of DNA replication and transcription, and induces apoptosis. However, treatment with cisplatin has been linked to the development of drug resistance and numerous undesirable side effects (e.g., severe kidney problems, myelosuppression, allergic reactions, reduced immunity to infections, peripheral neuropathy, gastrointestinal distress, hemorrhage, and tinnitus or hearing impairment) [6]. These side effects, particularly nephrotoxicity, are important factors limiting the efficacy of cisplatin in cancer therapy. Elucidation of the mechanism underlying cisplatin-induced nephrotoxicity may assist in protecting the kidneys and reducing the toxicity of cisplatin [7]. Following its entry into renal tubular cells, cisplatin activates signaling pathways such as sirtuin 1 (SITR1), mitogen-activated protein kinase (MAPK), p53, and reactive oxygen species (ROS), leading to cell death [8]. Moreover, it induces the production of tumor necrosis factor alpha (TNF-α), which aggravates the inflammatory response and accelerates the necrosis of renal tubular cells. In addition, cisplatin can also cause ischemic necrosis of the renal vascular structure, which results in further deterioration of renal function. These effects lead to the clinical development of AKI. Oxidative stress is the one of most important mechanisms involved in cisplatin toxicity. Besides DNA damage, it triggers cell death by apoptosis. In addition, the apoptotic pathways (extrinsic and intrinsic) involved in this process differ with the type of cancer [9].

Stem cells are undifferentiated primitive cells in the human body that can independently divide and proliferate, as well as differentiate into a variety of cells with specific functions. Stem cells from the apical papilla (SCAPs) are a type of mesenchymal stem cells (MSCs) [10]. SCAPs in the root apex of adolescent permanent teeth are characterized by a high potential for proliferation, self-renewal capacity, and low immunogenicity [11]. At present, only stem cells from human exfoliated deciduous teeth are used to treat kidney damage [12]. However, recent studies have found that stem cells can repair damaged tissues through paracrine and anti-inflammatory mechanisms [13]. Following damage to local tissue, MSCs are activated and secrete a variety of cytokines, forming a local microenvironment that is conducive to tissue repair. The secretory function of stem cells gives rise to the concept of cell-free therapy. Compared with stem-cell transplantation, cell secretions can avoid problems such as tumor formation and immune rejection.

Exosomes are lipid bilayer vesicles, 30–150 nm in size, that are secreted by the cell when the multivesicular bodies in the cell fuse with its plasma membrane. Exosomes are equivalent to the cytoplasm enclosed in a lipid bilayer and are rich in nucleic acids (such as microRNA, long noncoding RNA, circular RNA, messenger RNA, and transfer RNA), protein, cholesterol, and other typical cytoplasm contents [14,15]. The characteristic surface proteins of exosomes generally include CD63, CD81, CD9, tumor susceptibility 101 (TSG101), and heat shock 27 kDa protein 1 (HSP27). In terms of their biological function, exosomes can deliver various receptors, proteins, genetic material such as DNA and microRNA, and lipids to target cells. The target cells incorporate these rich inputs from exosomes in three main ways: receptor–ligand interaction, direct fusion with the plasma membrane, and endocytosis, which allows the exosome contents to protect and heal the cell [16]. An increasing number of studies have shown that the exosomes of MSCs exert a protective effect against AKI [17]. In a model of ischemia-/reperfusion-induced AKI, adipose MSC-derived exosomes offered protection from the AKI [18]. In addition to AKI, the therapeutic efficacy and protective mechanism of exosomes in various kidney diseases/disorders have also been explored, including lupus nephritis (LN), other glomerular diseases, diabetic nephropathy (DN), polycystic kidney disease (PKD), renal fibrosis and chronic kidney disease (CKD) [16,19,20]. Therefore, the aim of this study was to investigate the therapeutic effect of pretreatment with SCAP-derived exosomes (SCAP-ex) against AKI.

SCAP-ex were obtained by ultracentrifugation and identified via Western blotting. NRK-52E cells were pretreated with SCAP-ex, followed by induction of nephrotoxicity by treatment with cisplatin. The results confirmed the protective effect of SCAP-ex against inflammation, oxidative stress, and apoptosis.

## 2. Results

### 2.1. Characterization of SCAPs and SCAP-ex

SCAPs were isolated via enzyme digestion. The subcultured cells obtained from a single colony after isolation were gradually cultivated as an adherent monolayer and had a fibroblast-like morphology (Figure 1A). These SCAPs exhibited positive expression of the important MSC markers CD90 (99.78%) and CD73 (99.55%), as well as the endothelial progenitor marker CD105 (79.63%). However, they expressed minimal levels of the hematopoietic markers CD34 (0.09%) and CD45 (0.2%) (Figure 1B). 

Western blotting was used to analyze the expression levels of the characteristic proteins in SCAP-derived exosomes. As shown in Figure 2A, SCAP-ex expressed the proteins characteristic of exosomes, such as CD63 and CD81. Figure 2B shows the morphology and size of exosomes, as assessed using TEM. The structure of the exosomes was round (diameter: 30–150 nm), which is consistent with the typical structure of exosomes. The particle size distribution of SCAP-ex was analyzed via NTA. As shown in Figure 2C, the median particle size was 147 nm, the average was 168.2 nm, and the mode was 116.3 nm. These results show that SCAP-ex, with respect to marker proteins, morphology, and particle size, show characteristics typical of all exosomes.

### 2.2. SCAP-ex Protected Cisplatin-Treated NRK-52E Cells

The results of the MTT analysis showed that the survival of NRK-52E cells treated with SCAP-ex at different concentrations was close to 100% (Figure 3A), suggesting that SCAP-ex are nontoxic to NRK-52E cells. Additionally, treatment with cisplatin under different concentrations reduced the survival of NRK-52E cells in a dose-dependent manner (Figure 3B). After treatment with 15 μM cisplatin, approximately 60–80% of NRK-52E cells survived; hence, this concentration was used for subsequent experiments. NRK-52E cells were pretreated with 40 or 80 μg/mL SCAP-ex or 10 μM losartan, and subsequently treated with 15 μM cisplatin. The cell survival rate increased significantly (Figure 3C), and 80 μg/mL SCAP-ex exerted the strongest preventive and protective effects against cisplatin-induced injury, as was the case for losartan. Thus, a concentration of 80 μg/mL SCAP-ex was used in subsequent experiments.

### 2.3. SCAP-ex Promoted Cellular Vitality in Cisplatin-Treated NRK-52E Cells

In the cell-vitality assay we investigated the levels of free thiols by determining apoptosis and cell survival rates. Normal cells and early/late apoptotic cells exhibit different levels of fluorescence in response to the VitaBright-48 reagent, as shown in Figure 4A. Normal and early apoptotic cells are located in the lower right and left corners of the image, respectively. In the late stages of apoptosis, the cell membrane is severely damaged; thus, the PI reagent can enter the nucleus and emit fluorescence there. Therefore, late apoptotic cells can be distinguished using this reagent, and they can be seen in the upper area of the image. According to the quantitative results shown in Figure 4B, after pretreatment with SCAP-ex, the proportion of cisplatin-treated NRK-52E cells that were normal increased from 82.0% to 92.9%, while that of the early and late apoptotic cells decreased from 18.0% to 7.1%. Following pretreatment with losartan, the proportion of normal cells increased from 82.0% to 91.4%, while that of the early and late apoptotic cells decreased from 18.0% to 8.6%. These results revealed that SCAP-ex reduces the cell apoptosis caused by cisplatin, and its protective effect is similar to that of the clinical drug losartan. 

### 2.4. SCAP-ex Improved ROS/GSH/MDA/TNF-α Expression in Cisplatin-Treated NRK-52E Cells

Oxidative stress is an important mechanism in AKI. During acute cellular injury, mitochondria release excess ROS, increasing cell damage. After pretreatment of NRK-52E cells with SCAP-ex, the ROS levels in cisplatin-treated NRK-52E cells decreased from 176.4% to 123.2%. Following pretreatment with losartan, the ROS levels decreased from 176.4% to 114.1% (Figure 5A). These results showed that SCAP-ex can reduce the ROS content of cisplatin-treated NRK-52E cells and thus retard oxidative stress in cells.

Using the enzyme-cycle method, we used GSH reductase to quantify GSH, with the calibration curve *y* = 0.0789*x* + 0.3491 (*R*^2^ = 0.984). After pretreatment with SCAP-ex, the levels of GSH in cisplatin-treated NRK-52E cells increased from 9.3 to 16.27 μmol/g protein (Figure 5B). Pretreatment with losartan yielded similar results and increased the levels of GSH. 

MDA and thiobarbituric acid (TBA) react at high temperature (90–100 °C) and under acidic conditions to form the MDA-TBA complex, which can be measured using the colorimetric method at 530–540 nm wavelength, using the calibration curve *y* = 0.0005*x* + 0.0593 (*R*^2^ = 0.9879). After pretreatment with SCAP-ex, the levels of MDA in cisplatin-treated NRK-52E cells decreased significantly from 3.36 to 1.94 μmol/g protein (Figure 5C). Pretreatment with losartan yielded similar results.

We performed ELISA at 450 nm to quantitatively measure the TNF-α content of the cells. Following pretreatment of NRK-52E cells with SCAP-ex, the levels of TNF-α in cisplatin-treated NRK-52E cells decreased from 22.13 to 20.31 pg/mL (Figure 5D). Pretreatment with losartan exerted a similar effect.

### 2.5. Quantitative RT-PCR Assay of Cisplatin-Treated NRK-52E Cells Pretreated with SCAP-ex

Genes whose expression changed in response to damage caused by cisplatin-induced inflammation and apoptosis in NRK-52E cells included nuclear factor-κβ (NF-κβ), interleukin-1β (IL-1β), B-cell lymphoma-2 (Bcl-2), Bcl-2 associated X (Bax), p53, caspase-8 (CASP8), CASP9, and CASP3. Following pretreatment of NRK-52E cells with SCAP-ex and induction by cisplatin, the fold change of NF-κβ and IL-1β in their expression levels in response to the addition of cisplatin decreased from 4.61 to 2.07 and from 3.68 to 1.28, respectively. Pretreatment with losartan resulted in only slight decreases in expression (Figure 6A,B). The fold change in the gene-expression level of Bcl-2 increased from 1.47 to 3.81, whereas that of Bax decreased from 1.84 to 1.23; of p53 decreased from 2.47 to 0.8; and of CASP8, CASP9, and CASP3 decreased from 2.54 to 0.61, from 2.16 to 1.09, and from 2.38 to 1.36, respectively. Pretreatment with losartan resulted in weaker effects than pretreatment with SCAP-ex (Figure 7A–F).

## 3. Discussion

The therapeutic concept of cell-free therapy is based on the secretory function of stem cells. Compared with stem-cell transplantation, treatment with cell secretions can greatly reduce the risk of the tumor development, immune rejection, and ethical concerns [21,22]. SCAPs are a newly discovered type of MSCs that reside in the apical papilla of immature permanent teeth. When separated from the tip of the root and minced, they are found to contain the MSCs-associated positive markers CD73, CD90, CD105, and negative markers CD34 and CD45, indicating that they are not of hematopoietic origin. Therefore, SCAPs comprise a unique undifferentiated stem-cell lineage and are characterized by a high proliferative potential, self-renewal capacity, and low immunogenicity [13]. Based on our Western blotting analysis, SCAP-ex expressed the characteristic exosomal proteins CD63 and CD81. We observed the morphology and structure of SCAP-ex under TEM and found them to be round and cup-shaped, with a fingerprint-like membrane structure. NTA analysis and TEM observation showed that the particle size of these SCAP-ex was consistent with typical exosomes (i.e., 30–150 nm) [23]. These results show that isolated SCAP-ex may be a novel therapeutic agent for endodontics and other regenerative-medicine applications [24,25].

Previous studies have demonstrated that stem cells from different sources or derived exosomes exert therapeutic effects against cisplatin-induced AKI [17,26]. However, most are derived from bone marrow or umbilical-cord blood and are difficult to obtain; in contrast, it is relatively easy to obtain dental-cusp stem cells. At present, SCAP-ex are only used in dentine-pulp complex regeneration [25] and craniofacial soft-tissue regeneration [27]. Their therapeutic effect on AKI induced by cisplatin is unknown.

The results of the MTT assay showed that the viability of NRK-52E cells pretreated with SCAP-ex at different concentrations was close to 100%. These findings demonstrated that these exosomes were not toxic to the cells. Treatment with different concentrations of cisplatin reduced cell viability in a dose-dependent manner. After pretreatment with SCAP-ex or losartan, however, cell viability following cisplatin treatment was significantly greater, demonstrating that SCAP-ex exerts protective effects on the proliferation of cisplatin-treated NRK-52E cells.

The results of this study demonstrate that pretreatment with SCAP-ex could reduce cisplatin-induced ROS levels by 53.2%. Moreover, the levels of GSH increased by 78%, those of MDA decreased by 35%, and those of TNF-α decreased by 9%. These effects reduced the cisplatin-induced oxidative stress. Oxidative stress promotes cisplatin-induced nephrotoxicity via the accumulation of intracellular ROS [28]. This enhancement of the cells’ antioxidative capacity may be the underlying mechanism through which SCAP-ex inhibited cisplatin-induced renal cell apoptosis. The rates of early and late cell apoptosis were reduced by 0.4% and 10.6%, respectively, whereas overall cell viability was increased by 10.9%. The improvement in cell apoptosis was better than that observed after pretreatment with losartan. Cisplatin also increases the concentration of TNF-α (a pleiotropic cytokine with endocrine, paracrine, and autocrine proinflammatory effects) and affects IL-1β and NF-κβ [29]. After pretreatment with SCAP-ex, the expression levels of the inflammation-related factors NF-κβ and IL-1β in cisplatin-induced NRK-52E cells decreased by 52.9% and 76.3%, respectively. The antiapoptotic gene Bcl-2 was upregulated, whereas the proapoptotic genes Bax, p53, CASP8, CASP9, and CASP3 were downregulated. Because of the imbalance of Bcl-2 and Bax in the mitochondrial membrane, cytochrome c is released. This stimulates the downstream production of CASP9 and CASP3, and eventually induces apoptosis [30,31,32]. The pathway of inflammation and apoptosis caused by cisplatin revealed by our results is shown in Figure 8. Following entry into cells, cisplatin causes DNA damage, generates ROS, NF-κβ, IL-1β, and other factors, causing inflammation, inducing the production of p53, and stimulating Bax/Bcl-2 to act on mitochondria. These effects result in the production of CASP9 and CASP3, leading to cell apoptosis. TNF-α factor and CASP8 can also lead to apoptosis. Stem cells are used to treat damaged cells through paracrine therapy, and exosomal cell-free therapy may ameliorate cisplatin-induced nephrotoxicity through the inhibition of oxidative stress, the inflammatory response, and apoptosis.

## 4. Materials and Methods

### 4.1. Isolation and Identification of SCAPs

Apical papilla tissue was separated from the tip of the root following a standard operation procedure approved by the institutional review board of the Dental Clinic of Kaohsiung Medical University of Taiwan (KMUHIRB-SV(I)-20210047). SCAP cells were separated from apical papilla tissue using enzymatic digestion in a solution of 3 mg/mL collagenase type I (Worthington Biochemical, Lakewood, NJ, USA) and 4 mg/mL dispase (Sigma-Aldrich, St. Louis, MO, USA) for 1 h at 37 °C. Single-cell suspensions were obtained by passing digested samples through a strainer (70 μm) (Falcon; Thermo Fisher Scientific, Waltham, MA, USA). Cell suspensions were centrifuged at 1000 rpm for 10 min, and single cells were resuspended in culture medium composed of α-Minimal Essential Medium (Gibco; Thermo Fisher Scientific) with 10% fetal bovine serum and 1% antibiotic–antimycotic solution (Gibco; Thermo Fisher Scientific). Subsequently, the cells were incubated at 37 °C with 5% CO_2_. Subculturing was performed for ordinary cultures, and the medium was changed once every 2 d. The identification of isolated SCAPs was based on the presence of cell-surface molecules such as CD34, CD45, CD73, CD90, and CD105 and was performed using a FACSCalibur flow cytometer (BD Bioscience, Massachusetts, MA, USA).

### 4.2. Isolation and Identification of SCAP-ex

When the SCAP cells reached 80% confluence, they were cultured in serum-free medium containing 1% bovine serum albumin for 2 d. The medium supernatant was collected and centrifuged at 4 °C at 2000× *g* for 10 min and then 12,000× *g* for 30 min to eliminate dead cells and large-cell debris. The supernatant was filtered with a 0.22 μm filter and then ultracentrifuged at 100,000× *g* for 70 min (L-90K, Beckman, Indianapolis, IN, USA). The pellet was washed with 1 mL of phosphate-buffered saline (PBS) and again centrifuged at 100,000× *g* for 70 min (MAX-E, Backman, USA). The precipitate was suspended in PBS and quantified using the bicinchoninic acid (BCA) protein assay. The presence of proteins characteristic of SCAP-ex was examined using Western blotting. The morphometry of the exosomes was observed via transmission electron microscopy (TEM, FEI Tecnai, G2 F20 S-TWIN, Bellaterra, Spain). Finally, the size of the exosomes was measured via nanoparticle tracking analysis (NTA, NanoSight LM 10-HS, Malvern Panalytical, Malvern, UK).

### 4.3. Cellular Viability and Protection following SCAP-ex/Cisplatin Treatment

NRK-52E cells were purchased from the Bioresource Collection and Research Center (BCRC, Hsinchu, Taiwan). Cells were cultured as monolayers in Dulbecco’s modified Eagle’s medium containing 5% bovine calf serum and 1% penicillin/streptomycin in a humidified incubator with 5% CO_2_ at 37 °C. The culture medium was replaced every 2–3 d.

We sought to assess the cytotoxic effect of cisplatin or SCAP-ex on NRK-52E cells using the 3-(4,5-dimethylthiazol-2-yl)-2,5-diphenyltetrazolium bromide (MTT) assay. This is a colorimetric assay was based on MTT (Sigma-Aldrich, Missouri, MO, USA) for assessing cell metabolic activity as an indicator for cell viability, proliferation, and cytotoxicity. For this purpose, 1 × 10^5^ cells/well were seeded in 96-well plates containing culture medium with different concentrations of cisplatin or SCAP-ex. Cells in each well were treated with 5 mg/mL MTT at 37 °C for 4 h. The medium was removed and the formazan was solubilized in dimethyl sulfoxide. The metabolized MTT was measured based on optical density at 570 nm using a spectrophotometer (Multiskan FC, Thermo Fisher Scientific, Massachusetts, MA, USA). To evaluate the protective effect of SCAP-ex, NRK-52E cells were pretreated with 40 or 80 μg/mL SCAP-ex or 10 μM losartan for 30 min. Next, 0, 5, 10, or 15 μM cisplatin was added, and the cells were incubated for another 24 h, after which an MTT assay was performed.

### 4.4. Cellular Vitality Assay

NRK-52E cells (1 × 10^5^ cells/well) were pretreated with 80 μg/mL SCAP-ex or 10 μM losartan in 24-well plates for 30 min. Next, 15 μM cisplatin was added and the cells were cultured for another 24 h. The cultured cells were then centrifuged at 1500 rpm for 5 min. The supernatant was removed and the precipitate was suspended in 200 μL of PBS. The suspension was mixed continuously while 800 μL of cold ethanol was added, and the cells were stored at −20 °C overnight. The thiol levels were measured using VitaBright-48 (VB48, ChemoMetec A/S, Lillerod, Denmark) and the number of dead cells was determined using propidium iodide (PI, ChemoMetec A/S, Denmark), according to the instructions provided by the manufacturer of the cell-vitality kit (ChemoMetec A/S, Denmark). The results were obtained using a NucleoCounter NC-250 and the NucleoView software (ChemoMetec A/S, Denmark).

### 4.5. ROS Assay

NRK-52E cells (1 × 10^5^ cells/well) were pretreated with SCAP-ex or losartan, followed treated with cisplatin, as Section 2.4 process. The medium was removed, and the cultured cells were rinsed with PBS. Dichlorofluorescin diacetate (DCFDA, 100 μL) was added to each well, and the cells were incubated at 37 °C for 45 min in the dark. Subsequently, the DCFDA solution was removed, and the cells were rinsed twice with 100 μL of PBS per well. The samples were placed on a fluorescent enzyme-linked immunosorbent assay (ELISA) reader (Multiskan FC, Thermo Fisher Scientific, Massachusetts, MA, USA) to evaluate excitation/emission at 485/535 nm.

### 4.6. Glutathione (GSH) and Malondialdehyde (MDA) Assay

NRK-52E cells (1.5 × 10^5^ cells/well) were pretreated with SCAP-ex or losartan, followed treated with cisplatin, as in the Section 2.4 process. The medium was removed, and the cultured cells were rinsed twice with PBS. The cells were lysed, deproteinized, and centrifuged at 10,000× *g* for 15 min at 4 °C. The supernatants were collected for the assay. To determine the activity of GSH S-transferase, 50 μL of sample and standard solution were added to a 96-well plate. After the addition of 150 μL of assay cocktail and shaking for 25 min in the dark, the absorbance at 405 nm was measured using an ELISA reader (Multiskan FC, Thermo Fisher Scientific, USA). 

For the MDA assay, cell lysates were centrifuged at 10,000× *g* for 15 min at 4 °C. Whole homogenates were collected for the assay. Sample and standard solution (100 μL) were added to an Eppendorf tube along with 100 μL of trichloroacetic acid (10%) assay reagent, and the solution was mixed well. Next, 800 μL color reagent was added, and the Eppendorf tube was placed in boiling water for 1 h. After 10 min of incubation on ice, the Eppendorf tubes were centrifuged at 1600× *g* at 4 °C. Each sample and standard was loaded into 96-well assay plates. The absorbance of each well was measured at 540 nm using an ELISA reader (Multiskan FC, Thermo Fisher Scientific, USA). MDA concentrations were calculated according to the instructions provided by the manufacturer (Cayman Chemical, Ann Arbor, MI, USA).

### 4.7. TNF-α Assay

NRK-52E cells (1 × 10^4^ cells/well) were pretreated with SCAP-ex or losartan, followed treated with cisplatin, as in the Section 2.4 process. The culture medium was collected to determine the concentration of TNF-α secreted from NRK-52E cells using an ELISA MAX Deluxe set kit (BioLegend, San Diego, CA, USA). The absorbance at 450 nm was measured using a microplate reader and the concentrations were determined using standard curves.

### 4.8. Reverse Transcription–Polymerase Chain Reaction (RT-PCR)

NRK-52E cells (5 × 10^5^ cells/well) were pretreated with SCAP-ex or losartan, followed treated with cisplatin, as in the Section 2.4 process. The total RNA from the NRK-52E cells was extracted using Trizol reagent (Ambion^®^, Life Technologies™, Carlsbad, CA, USA) for 10 min, and the RNA quantity was determined using a NanoDrop 2000 spectrophotometer (Thermo Fisher Scientific, USA). Complementary DNA (cDNA) was synthesized from 1000 ng of RNA using the iScript cDNA synthesis kit (Bio-Rad, Hercules, CA, USA) and a thermocycler (5 min at 25 °C, 20 min at 46 °C, 1 min at 95 °C). RT-PCR was performed using the iQuant SYBR Green Supermix (Bio-Rad, Hercules, CA, USA) according to the instructions provided by the manufacturer. Initial denaturation was performed at 95 °C for 15 min, followed by 60 cycles of 95 °C for 15 s and 60 °C for 60 s. The relative gene-expression fold change was determined using the 2^−∆∆CT^ method; the levels were normalized to those of the housekeeper gene glyceraldehyde-3-phosphate dehydrogenase (GAPDH). The used PCR primers were as follows: Bax (Forward: 5′-TGG AGC TGC AGA GGA TGA TTG-3′, Reverse: 5′-GGT CCC GAA GTA GGA AAG GAG-3′); Bcl-2 (Forward: 5′-GAA CTG GGG GAG GAT TGT GG-3′, Reverse: 5′-GAA CTG GGG GAG GAT TGT GG-3′); IL-1β (Forward: 5′-ATA GCA GCT TTC GAC AGT GAG G-3′, Reverse: 5′-CAA TCC TTA ATC TTT TGG GGT CTG T-3′); NF-κβ (Forward: 5′-CGC TTC TCA GGA GTT CCA GCT AT-3′, Reverse: 5′-GGG ATG TCG GCA GCA TTG AT-3′); p53 (Forward: 5′-CCC AGG GAG TGC AAA GAG AG-3′, Reverse: 5′-GGT CTT CGG GTA GCT GGA GT-3′); caspase3 (Forward: 5′-GGA GCT TGG AAC GCG AAG A-3′, Reverse: 5′-TCT CAA TAC CGC AGT CCA GC-3′); caspase8 (Forward: 5′-ACT GCA AGA CAA CTC GAG CC-3′, Reverse: 5′-TCC TCA CCT CGA GGA CAT CT-3′); caspase9 (Forward: 5′-TAC TCC AGG GAA GAT CGA GAG ACA-3′, Reverse: 5′-AGC CGT GAC CAT TTT CTT AGC AG-3′); GAPDH (Forward: 5′-ACC ATC TTC CAG GAG CGA GA-3′, Reverse: 5′-GGT GGT GAA GAC GCC AGT AG-3′).

### 4.9. Statistical Analysis

All experiments were performed thrice for different samples. All data are presented as means ± standard deviation (SD) and performed using IBM SPSS Statistics Base 30U software through ANOVA analysis. Statistical comparisons were performed, and *p* values smaller than 0.05 were considered significant.

## 5. Conclusions

In summary, the results of this study indicated that SCAP-ex may protect NRK-52E cells from cisplatin-induced AKI by inhibiting oxidative stress, inflammation, and cell apoptosis. SCAPs are neural-crest-derived MSCs, and SCAP-ex contains numerous bioactive compounds that are key factors in stem-cell paracrine action. SCAP-ex, rather than MSCs, may thus be useful as a cell-free therapeutic strategy against AKI induced by chemotherapeutic agents.

## Figures and Tables

**Figure 1 ijms-23-05721-f001:**
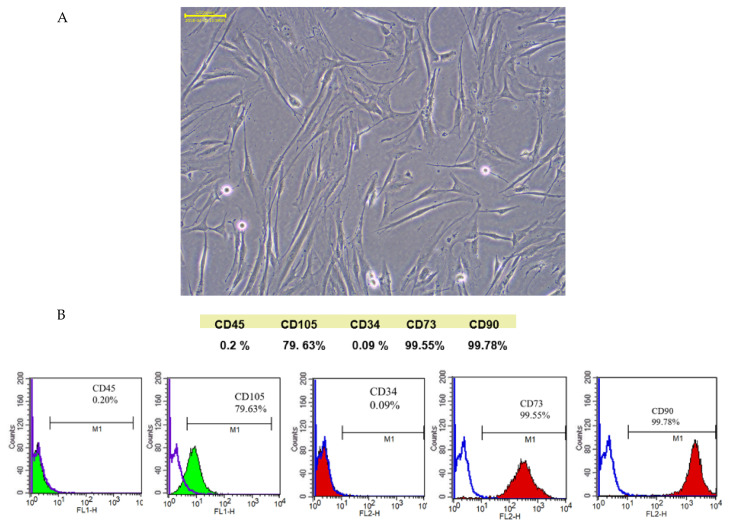
Morphology; scale bar is 0.1 mm (**A**) and cellular markers of SCAPs (**B**). SCAPs, stem cells from the apical papilla.

**Figure 2 ijms-23-05721-f002:**
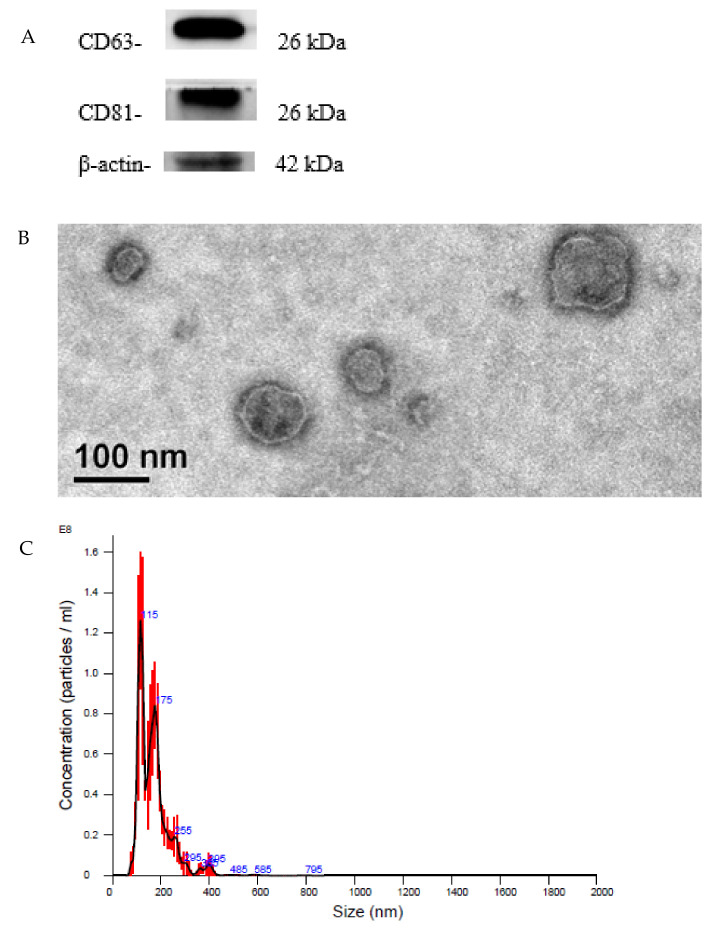
Characteristic proteins of SCAP-isolated exosomes (**A**). Morphology of SCAP-ex under TEM (**B**). Size distribution of SCAP-ex by NTA (**C**). NTA, nanoparticle tracking analysis; SCAPs, stem cells from the apical papilla; SCAP-ex, SCAP-derived exosomes; TEM, transmission electron microscopy.

**Figure 3 ijms-23-05721-f003:**
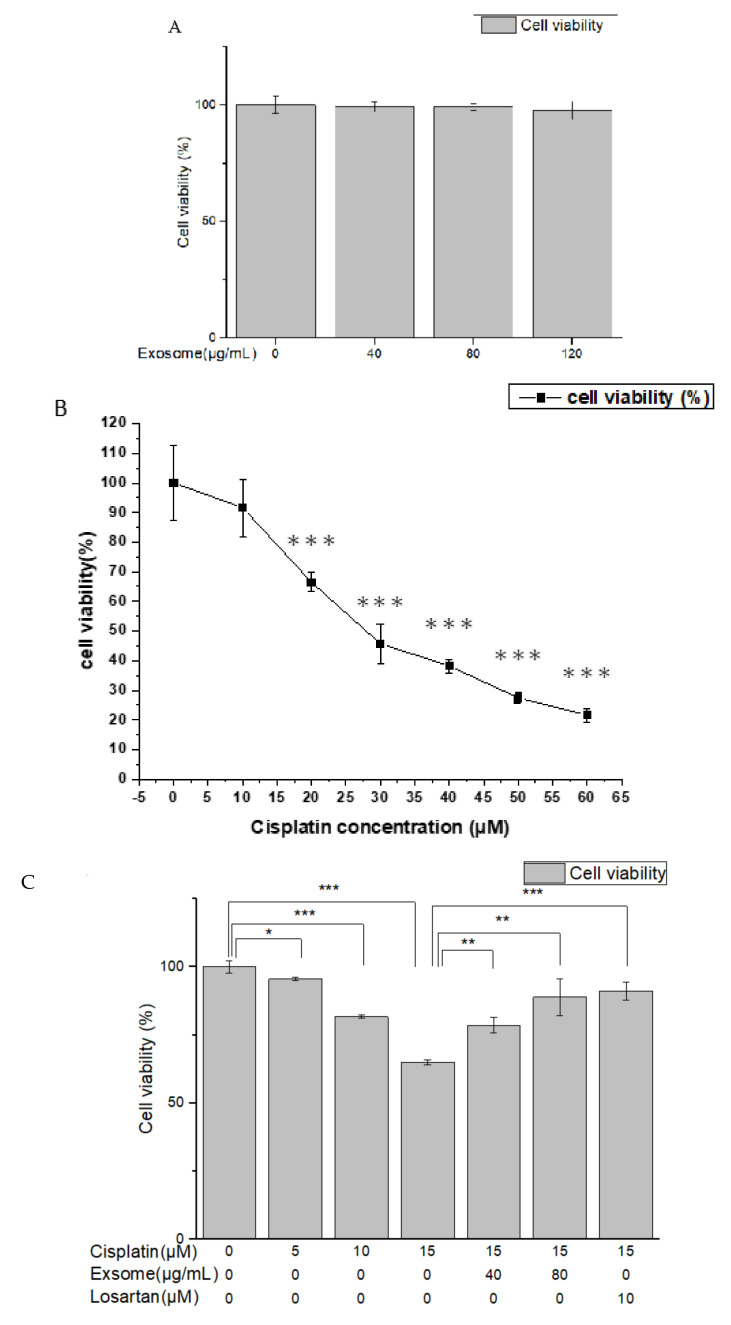
Cell viability of NRK-52E cells treated with various concentrations of SCAP-ex (**A**) and cisplatin (**B**). Cell survival of NRK-52E cells pretreated with 40 or 80 μg/mL SCPA-ex or 10 μM losartan and subsequently treated with 15 μM cisplatin (**C**). Values are expressed as mean ± SD (*n* = 3), * *p* < 0.05; ** *p* < 0.01; *** *p* < 0.001. SCAPs, stem cells from the apical papilla; SCAP-ex, SCAP-derived exosomes.

**Figure 4 ijms-23-05721-f004:**
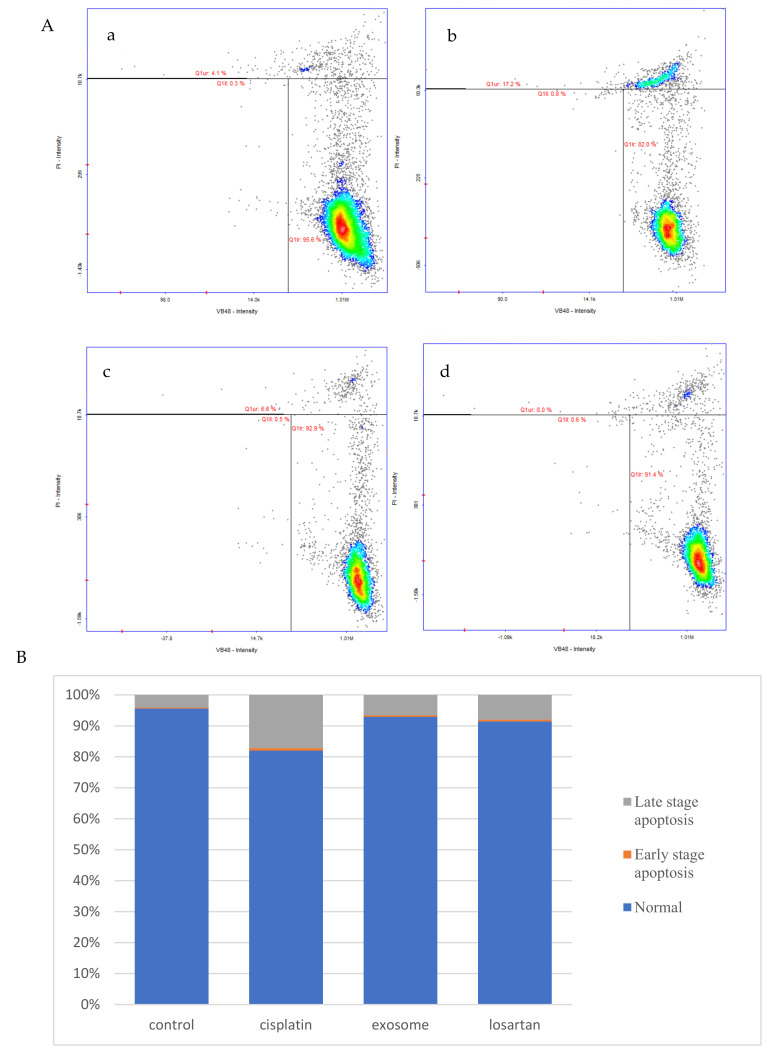
Distribution of normal and early/late apoptotic NRK-52E cells treated with or without cisplatin, SCPA-ex, and/or losartan (**A**), (**a**) control group, (**b**) cisplatin group, (**c**) exosome group and (**d**) losartan group; the lower right is normal cells, the lower left and upper sides of panels represent the early and late stages of apoptosis, respectively. Percentage of normal, early, and late apoptotic NRK-52E cells treated with or without cisplatin, SCPA-ex, and/or losartan (**B**). SCAPs, stem cells from the apical papilla; SCAP-ex, SCAP-derived exosomes.

**Figure 5 ijms-23-05721-f005:**
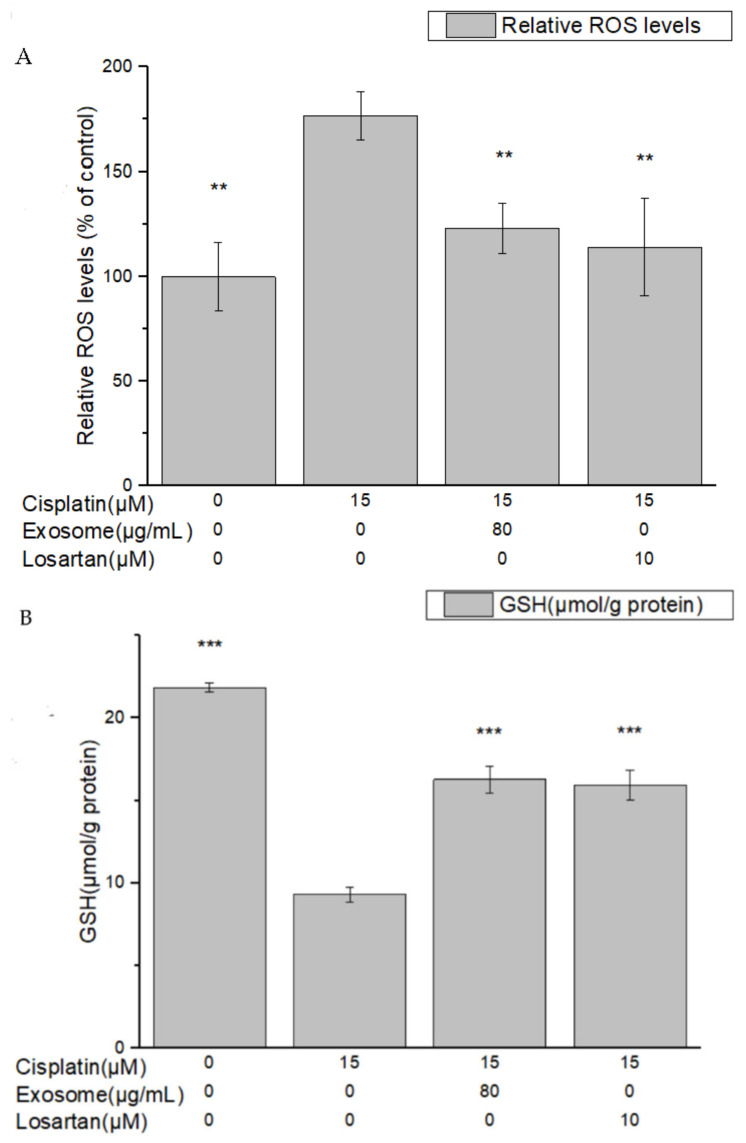
ROS expression (**A**), GSH levels (**B**), MDA levels (**C**) and TNF-α levels (**D**) in cisplatin-treated NRK-52E cells after pretreatment with SCAP-ex or losartan. Values are expressed as mean ± SD (*n* = 3), * *p* < 0.05; ** *p* < 0.01; *** *p* < 0.001 for compared with cisplatin treated group. GSH, glutathione; MDA, malondialdehyde; ROS, reactive oxygen species; SCAPs, stem cells from the apical papilla; SCAP-ex, SCAP-derived exosomes; TNF-α, tumor necrosis factor-alpha.

**Figure 6 ijms-23-05721-f006:**
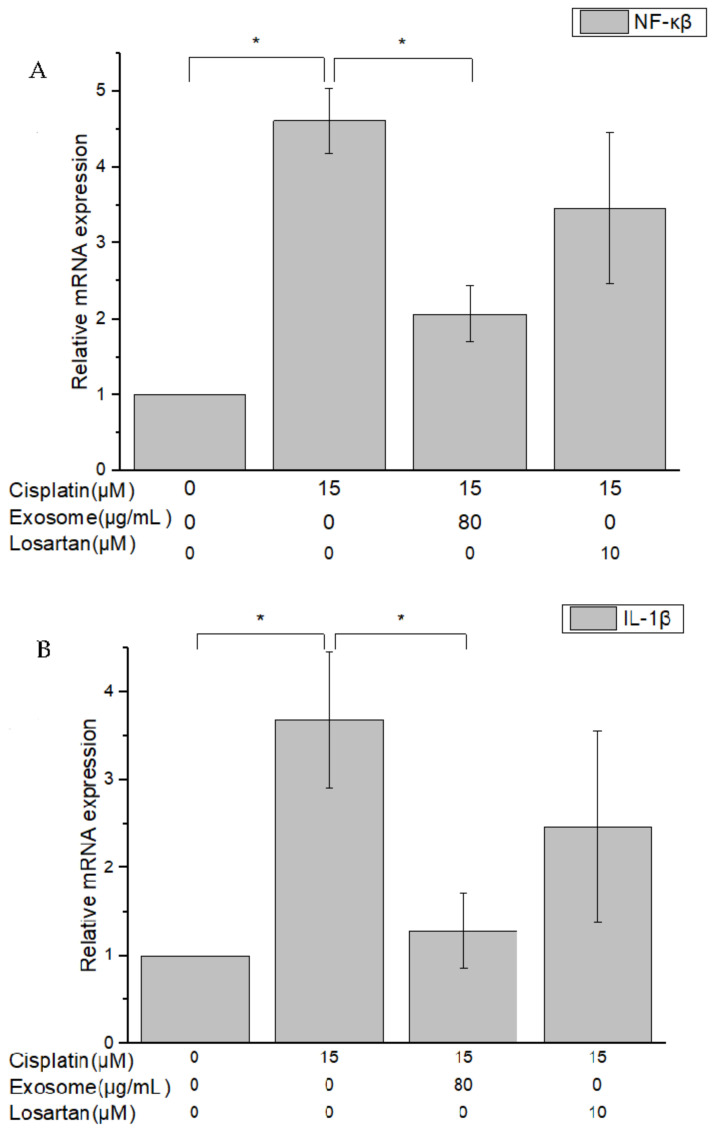
Gene-expression levels of NF-κβ (**A**) and IL-1β (**B**) in cisplatin-treated NRK-52E cells after pretreatment with SCAP-ex or losartan, measured via qRT-PCR assay. Values are expressed as mean ± SD (*n* = 3), * *p* < 0.05 for compared with cisplatin-treated group. IL-1β, interleukin-1β; NF-κβ, nuclear factor-κβ; qRT-PCR, quantitative reverse transcription–polymerase chain reaction; SCAPs, stem cells from the apical papilla; SCAP-ex, SCAP-derived exosomes.

**Figure 7 ijms-23-05721-f007:**
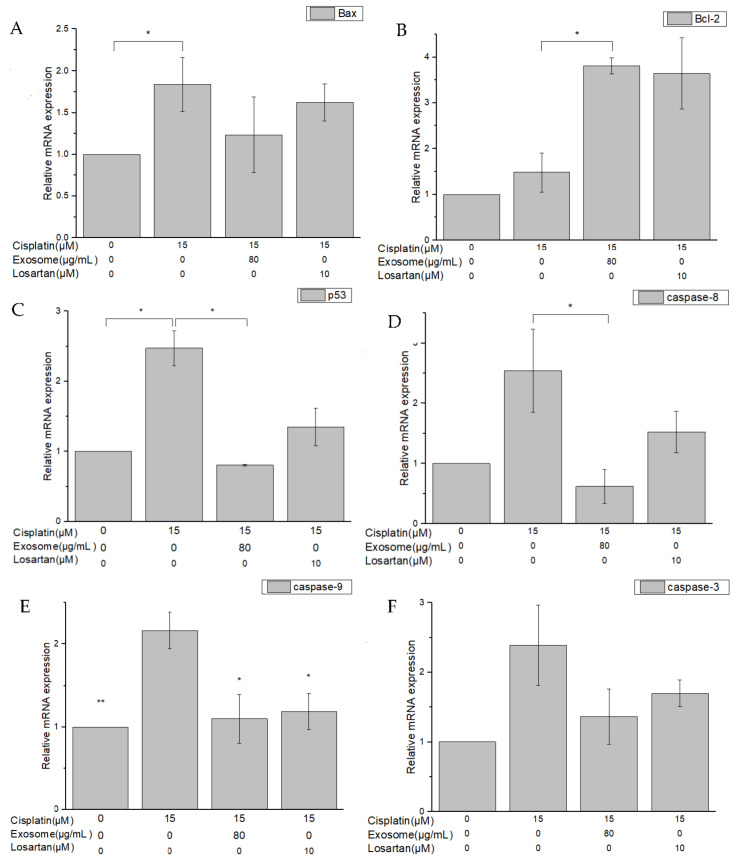
Gene-expression levels of Bax (**A**), Bcl-2 (**B**), p53 (**C**), caspase-8 (CASP8) (**D**), CASP9 (**E**), and CASP3 (**F**) in cisplatin-treated NRK-52E cells after pretreatment with SCAP-ex or losartan, measured via qRT-PCR assay. Values are expressed as mean ± SD (*n* = 3), * *p* < 0.05; ** *p* < 0.01 for compared with cisplatin-treated group. Bax, Bcl-2 associated X; Bcl-2, B-cell lymphoma-2; qRT-PCR, quantitative reverse transcription–polymerase chain reaction; SCAPs, stem cells from the apical papilla; SCAP-ex, SCAP-derived exosomes.

**Figure 8 ijms-23-05721-f008:**
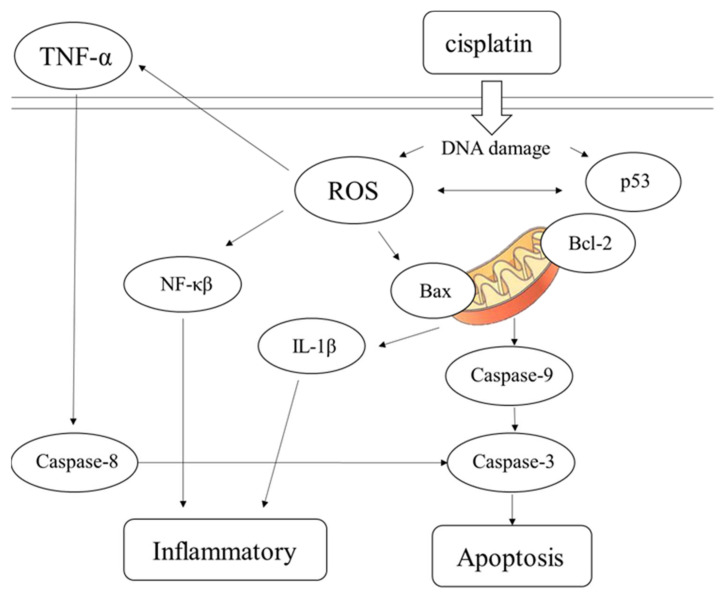
The speculated pathway of inflammation and apoptosis caused by cisplatin in NRK-52E cells.

## Data Availability

Not applicable.

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
