# Peer review of "Therapeutic Potential of Pretreatment with Exosomes Derived from Stem Cells from the Apical Papilla against Cisplatin-Induced Acute Kidney Injury"

_ijms, 2022, doi:10.3390/ijms23105721_

Round 1

Reviewer 1 Report

This study examines the ability of stem cells derived from apical papilla to produce exosomes with properties that enable them to combat cisplatin-induced injury to rat renal epithelial cells.  One interesting choice made in this study is the comparison of exosome effects to a long-used drug, losartan.

Losarten classification is not mentioned until the discussion.  This should be brought up in the introduction around line 45.   Also, on line 93, mention the what miR-199a-3p is thought to do and 2 lines below, what is downstream of 14-3-3?

In the Materials and Methods or at the beginning of the Results section tell us how you came to use losartan at 10 microM (source and/or reference that justifies this concentration) and was the choice made before the study began or during the study.

Discuss possible variation in exosome contents from SCAP lines.   Have you used more than one source for establishment of SCAP lines and did you find differences that might impact the outcome?  At least indicate possible differences in exosome contents between your exosomes and exosome from other stem cell lines, e.g. miR-199a-3p.  This might entail more characterization of the exosomes in your study.

What are your plans for more testing that would confirm lack of an immune response from using the exosomes-mouse model?

Author Response

Response to Reviewers

The authors would like to express my sincerely thanks to the editor and reviewers for the professional comments and constructive suggests. The article was revised following the suggestions. The modifications are listed as below.

Reviewer#1:

  1. Losarten classification is not mentioned until the discussion.  This should be brought up in the introduction around line 45.   Also, on line 93, mention the what miR-199a-3p is thought to do and 2 lines below, what is downstream of 14-3-3?

Response: We have added the losartan statement in the introduction around line 45, and rewriting the statement of line 93-95 to focused on the exosomes.

  1. In the Materials and Methods or at the beginning of the Results section tell us how you came to use losartan at 10 microM (source and/or reference that justifies this concentration) and was the choice made before the study began or during the study.

Response: After the concentration of cisplatin 15uM was selected, the cell viability was measured with different concentrations of losartan, and the protection effect of 10uM losartan was the best, so this concentration was selected.

  1. Discuss possible variation in exosome contents from SCAP lines.   Have you used more than one source for establishment of SCAP lines and did you find differences that might impact the outcome?  At least indicate possible differences in exosome contents between your exosomes and exosome from other stem cell lines, e.g. miR-199a-3p.  This might entail more characterization of the exosomes in your study.

Response: We did not compare the differences in exosome production or gene expression between different sources of SCAP, but we found that there were significant differences in the production of exosomes secreted by different stem cells between SHEDs and SCAP. The differences in the comments asked by the reviewers will be taken into consideration in future research.

  1. What are your plans for more testing that would confirm lack of an immune response from using the exosomes-mouse model?

Response: Before this study, the research on the therapeutic effect of SCAP exosomes on AKI was very limited. According to this research result, animal experiments on the protective effect of SCAP exosomes on AKI or the mechanism of autophagy behavior can be continued in the future. In addition, cells can be transfected with related genes such as miR-199a-3p to enhance or inhibit the expression of related genes and evaluate the therapeutic effect of the secreted exosomes.

Reviewer 2 Report

The authors investigated the effect of exosomes derived from stem cells from the apical papilla (SCAP-ex) on a cell model of acute kidney injury. In particular, the authors have pre-treated rat renal epithelial cells with SCAP-ex and then with cisplatin to induce acute injury. The results showed that the treatment with exosomes decreased oxidative stress, inflammation, and apoptosis (decreased pro-apoptotic factors and increased the anti-apoptotic ones).

The study is of clear design, the methodology is presented in sufficient detail, and the results are sufficiently described and well presented. However, there are points that the authors need to correct.

  1. Abstract, line 17; the exosomes were obtained by ultracentrifugation and not by “gradient centrifugation”. The methods do not mention the preparation of a density gradient.
  2. Line 18; please, change “oxidation stress” with “oxidative stress”.
  3. Lines 129-132; the method used to evaluate the morphometry of exosomes should be described.
  4. Line 140; “3-(4, 5-dimethylthiazol-2-yl)-2,5-diphenyltetrazolium bromide” was already abbreviated in the line above.
  5. Methods: the sentences “NRK-52E cells (1 × 105 cells/well) were pretreated with 80 μg/mL SCAP-ex or 10 μM 164 losartan in 24-well plates for 30 min. Next, 15 μM cisplatin was added and the cells were 165 cultured for another 24 h.” were repeated five times.
  6. Lines 298-299; it is unclear what the 4 (a, b, c, d) panels of figure 4A represent.
  7. Lines 339, 342, 343; the values 3.87, 1.71, and 1.69 are different from does shown in figures 6 and 7. Which are correct?
  8. Lines 386-388; it would be better to introduce these explanations at the end of the introduction to clarify the reasons why losartan was used.

Author Response

Response to Reviewers

The authors would like to express my sincerely thanks to the editor and reviewers for the professional comments and constructive suggests. The article was revised following the suggestions. The modifications are listed as below.

Reviewer#2:

  1. Abstract, line 17; the exosomes were obtained by ultracentrifugation and not by “gradient centrifugation”. The methods do not mention the preparation of a density gradient.

Response: We have revised to “differential ultracentrifugation”.

  1. Line 18; please, change “oxidation stress” with “oxidative stress”.

Response: We have revised to “oxidative stress”.

  1. Lines 129-132; the method used to evaluate the morphometry of exosomes should be described.

Response: We have revised to The morphometry of the exosomes was observed via transmission electron microscopy (TEM, FEI Tecnai, G2 F20 S-TWIN, Bellaterra Spain)”.

  1. Line 140; “3-(4, 5-dimethylthiazol-2-yl)-2,5-diphenyltetrazolium bromide” was already abbreviated in the line above.

Response:We have revised toThis is a colorimetric assay was based on MTT (Sigma-Aldrich, USA)”.

  1. Methods: the sentences “NRK-52E cells (1 × 105 cells/well) were pretreated with 80 μg/mL SCAP-ex or 10 μM 164 losartan in 24-well plates for 30 min. Next, 15 μM cisplatin was added and the cells were 165 cultured for another 24 h.” were repeated five times.

Response: We have revised to NRK-52E cells (1 × 105 cells/well) were pretreated with SCAP-ex or losartan, followed treated with cisplatin, as section 2.4 process”.

  1. Lines 298-299; it is unclear what the 4 (a, b, c, d) panels of figure 4A represent.

Response: We have added (a) control group, (b) cisplatin group, (c) exosome group and (d) losartan group in Fig. 4A statements.

  1. Lines 339, 342, 343; the values 3.87, 1.71, and 1.69 are different from does shown in figures 6 and 7. Which are correct?

Response: We have revised these wrong values to corrected values 4.61, 2.47 and 2.16.

  1. Lines 386-388; it would be better to introduce these explanations at the end of the introduction to clarify the reasons why losartan was used

Response: We have revised the statement to the end of the introduction, as reviewer’s suggestion.

Round 2

Reviewer 1 Report

Looks OK. Please, perform the the requested experiments in what will hopefully be soon forthcoming papers.